# Assessment of Nine Real-Time PCR Kits for African Swine Fever Virus Approved in Republic of Korea

**DOI:** 10.3390/v16101627

**Published:** 2024-10-17

**Authors:** Siwon Lee, Tae Uk Han, Jin-Ho Kim

**Affiliations:** 1R&D Team, LSLK Co. Ltd., Gimpo 10111, Republic of Korea; siwonlee99@nate.com; 2Waste-to-Energy Research Division, Environmental Resources Research Department, National Institute of Environmental Research, Incheon 22689, Republic of Korea; taeukhan@korea.kr; 3Institute of Tissue Regeneration Engineering (ITREN), Dankook University, Chungnam 31116, Republic of Korea; 4Department of Chemistry, College of Science and Engineering, Dankook University, Chungnam 31116, Republic of Korea

**Keywords:** African swine fever virus, approved ASFV diagnostic kits, food waste sample, monitoring system, real-time PCR

## Abstract

The African swine fever virus (ASFV) causes severe disease in wild and domestic pigs, with high mortality rates, extensive spread, and significant economic losses globally. Despite ongoing efforts, an effective vaccine remains elusive. Therefore, effective diagnostic methods are needed to rapidly detect and prevent the further spread of ASF. This study assessed nine commercial kits based on real-time polymerase chain reaction (PCR) approved in the Republic of Korea using the synthesized ASFV plasmid, 20 food waste samples, and artificially spiked samples (ASSs). The kits were evaluated for their diagnostic sensitivity, specificity, cost per reaction, and reaction running time. In addition, the results were compared with those of the World Organization for Animal Health (WOAH) standard methods. Three commercial kits (VDx^®^ ASFV qPCR Kit, Palm PCR™ ASFV Fast PCR Kit, and PowerChek™ ASFV Real-time PCR Detection Kit Ver.1.0) demonstrated the highest sensitivity (100 ag/μL), cost-effectiveness (less than KRW 10,000), and shortest running time (less than 70 min). These kits are suitable for the monitoring, early diagnosis, and prevention of the spread of ASF. This is the first report on the performance comparison of ASFV diagnostic kits approved in the Republic of Korea, providing valuable information for selecting kits for testing with food waste samples.

## 1. Introduction

African swine fever virus (ASFV) is an enveloped double-stranded DNA virus that contains domains such as the core shell, inner lipid envelope, capsid of the intracellular virion, and an additional external envelope [1,2,3]. Previous studies have attempted a classification of ASFV isolates based on the p72 gene (encoding the structural p72 protein), B602L (central variable region [CVR]), the E183L gene (encoding p54 protein), or EP402R (encoding for CD2V) [4,5,6,7]. Recently, ASFV has been classified into six groups, based upon the p72 gene by whole genome sequencing, because this gene remains a conserved region when significant changes have occurred in the genome [7]. ASFV causes ASF, a severe disease in wild and domestic pigs. In addition, ASF is associated with high mortality rates (100%), widespread outbreaks, and significant economic losses in affected regions [8,9]. Consequently, the World Organization for Animal Health (WOAH) has reported that ASF infectious diseases have occurred in domestic and wild pigs [10,11]. In addition, the risk of ASFV transmission through leftover and undercooked pork meat has been acknowledged, raising concerns about secondary infections from food waste used in livestock feed [12]. In particular, South Korea has limited the feeding of food waste to pigs through regulatory measures [12]. As mentioned, the proactive monitoring of ASFV in food waste and related facilities is essential. A few years ago, an ASF vaccine targeting multigene family (MGF) 360-505R and CD2v gene deletion emerged as an ASF vaccine candidate. However, an effective vaccine for ASF control has not been developed [13,14,15]. Therefore, highly sensitive and specific diagnostic assays are required to detect and control ASFV.

To diagnose ASFV, the WOAH manual suggests three polymerase chain reaction (PCR)-based methods: one conventional PCR and two real-time PCR with high sensitivity, specificity, and throughput for confirming and screening ASFV-suspected samples [16,17,18]. Since the WOAH-recommended standard methods were announced, advanced techniques such as ELISA, loop-mediated isothermal amplification (LAMP), duplex, triplex, and highly sensitive real-time PCR have been developed (Figure 1) [19,20,21,22]. However, most of these methods have been developed and assessed only for research purposes. Therefore, to facilitate their selection and use, commercial and licensed kits for ASFV should be evaluated by end users and countries. For example, Tignon et al. compared 12 commercial kits based on real-time PCR for ASFV detection, which were approved in Germany, with the WOAH-recommended method (2011) [23,24]. It has demonstrated that all the kits approved in Germany can detect ASFV in a wide range of sample types and provide information followed by compatibility and prioritization [23]. This study aims to assess nine ASFV detection kits licensed by the Ministry of Agriculture, Food, and Rural Affairs (MAFRA) in the Republic of Korea in terms of their performance using 20 food waste samples and a synthesized ASFV-positive control. The evaluation focuses on the specificity, sensitivity, cost per reaction, and reaction time, with the results of the final kit compared to those from a previous study [12].

## 2. Materials and Methods

This study assessed nine kits for detecting ASFV based on real-time PCR, licensed by MAFRA in the Republic of Korea, using 20 food waste samples with a synthesized ASFV-positive control. In a previous study conducted by the research team of the National Institute of Environmental Research (NIER) from 2020 to 2022, ASFV was analyzed in a total of 5581 samples collected from group catering, food waste disposal, and food waste treatment facilities, where the results were negative in all samples [25]. Based on the results, 20 food waste samples from 5581 samples were randomly selected that were used along with a synthesized ASFV positive control to enhance the significance of the experiments (e.g., analytical sensitivity and validation). All kits were evaluated for their analytical specificity, analytical sensitivity, cost per reaction, and reaction time. Additionally, the results of the final kit were compared with those of our previous study [12].

To prevent ASFV leakage in testing laboratories and farms, the ASFV isolate HBNH-2019 (National Center for Biotechnology Information [NCBI] accession number MN207061.1) sequence (1941 nucleotides [nt]) was synthesized (Magrogen Co. Ltd., Seoul, Republic of Korea). In addition, *porcine circovirus 2* (PCV2; NC_005148; 1034-1283, 250 nt), *porcine parvovirus* (PPV; NC_001718; 2387-2636, 250 nt), and *porcine pseudorabies virus* (PrV; NC_006151; 66,781-67,030, 250 nt) were synthesized as reference viruses. All the synthesized nucleic acid fragments were inserted into the pUC57 vector and used as positive controls, as well as three reference viruses for analytical specificity testing.

The nine approved ASFV diagnostic kits were purchased in the Republic of Korea. All kit information, such as the manufacturer’s name, kit name, number of tests, permit number, and permit date, is listed in Table 1. All PCRs were performed using the CFX connect™ Real-time System (Bio-Rad, Hercules, CA, USA). PCR was performed according to the manufacturer’s instructions. The reaction time and quantification cycle (Cq) values were determined using CFX Maestro software Ver. 3.0 (Bio-Rad, Hercules, CA, USA). HiGene™ RNase Free Water (BIOFACT Co., Ltd., Daejeon, Republic of Korea) was used as a negative control in the PCR.

DNA was extracted from all 20 samples using the methods described by Lee et al. [12]. Twenty food waste samples were obtained from pork meat-based foods collected from biogas and composting treatment facilities. The samples were stored in an ultra-low freezer. One gram of each sample was homogenized for 40 s at 6 m/s using MP FastPrep^®^ 24 (MP Biomedicals, Irvine, CA, USA) [26]. Ground sample solutions were used for DNA isolation using the QIAamp^®^ DNA mini kit (Qiagen, Hilden, Germany), according to the manufacturer’s instructions. Using isolated DNA samples as a diluent, the ASFV plasmid (1 ng/μL) was serially 10-fold diluted up to 10^−8^ (10 ag/μL) for analysis of the sensitivity, called artificially spiked samples (ASSs). Moreover, 100 pg/μL of the ASFV plasmid was used as a positive control in the PCR.

Two different relative fluorescence units (RFU) (1000 and 2500 RFU) were used for Cq analysis. If the Cq values were <40, most kits were considered positive. However, three kits have different Cq criteria for positivity: PowerChek™ ASFV Real-time PCR Detection Kit Ver. 1.0 0 (Kogene Biotech, Republic of Korea) is ≤38; Virotype ASFV Real-time PCR Kit (Indical Bioscience, Germany) is <35; VetMax™ ASFV Detection Kit (Thermofisher, MA, USA) is <45. Therefore, kit-specific Cq and identical Cq criteria (<35) were used to analyze and evaluate the real-time PCR results. In addition, three analysts repeated the specificity and analytical sensitivity tests of the selected kits, according to the methods described in this study for validation.

## 3. Results

The ASFV plasmids were positive, and the internal controls among the reference viruses were successfully distinguished using nine kits with 1000 RFU (Figure 2). However, the internal controls of four kits (Virotype ASFV Real-time PCR Kit, Indical Bioscience, Germany; ID Gene™ African Swine Fever Duplex, Innovative Diagnostics, France; Opti ASFV qPCR Kit, Optipharm, Republic of Korea; Genelix™ ASFV Real-time PCR Detection Kit, Sanigen, Republic of Korea) could not be identified with 2500 RFU. Additionally, the ASFV plasmid (100 ng/μL) was not detected by two kits (ID Gene™ African Swine Fever Duplex and Opti ASFV qPCR Kit).

Using twenty food waste samples, all nine kits yielded negative results. This finding is consistent with that of our previous study [12]. Consequently, ASSs were used to test the analytical sensitivity (Appendix A). As a result, most of the kits could detect less than 1 fg/μL of ASSs when these data were analyzed using 1000 RFU and the Cq criteria of the kit. Notably, 100 ag/μL of ASSs could be identified by three kits from Median Diagnostics, Ahram Biosystems, and MiCo Biosystems. Moreover, four kits from Median Diagnostics, Indical Bioscience, Ahram Biosystems, and MiCo BioMed could detect ASSs until 1 fg/μL when the Cq criteria was adjusted to <35. In addition, five kits from Median Diagnostics, Indical Bioscience, Ahram Biosystems, MiCo BioMed, and Kogene Biotech could detect 10 fg/μL of ASSs, even if the Cq criteria were strictly regulated <35.0 with 2500 RFU. However, the MiCo Biosystems kit is more expensive than the nine approved ASFV detection kits.

Regarding the reactivity, analytical specificity, and sensitivity, the VDx^®^ ASFV qPCR Kit (Median Diagnostics, Chuncheon-si, Republic of Korea) and the Palm PCR™ ASFV Fast PCR Kit (Ahram Biosystems, Seoul, Republic of Korea) were selected for validation by three analysts. The PowerChek™ ASFV Real-time PCR Detection Kit Ver.1.0 (Kogene Biotech, Seoul, Republic of Korea) was also included in the test because its analytical sensitivity was relatively similar to 10 fg/μL of ASSs when the data were analyzed with 2500 RFU and <35 C values. Therefore, the three kits were evaluated in the validation test, and all the validation data are presented in Table 2. Most analytical sensitivity results were 10 times lower than those reported by the developer. However, only the Kogene Biotech kit in 1000 RFU with <35 Cq values was 10 times more sensitive than the developer’s claim. In addition, the average detection limit of Kogene Biotech with 1000 RFU and kit-specific Cq values (≤38) was highest at 316.23 ag/μL of ASSs. When the analysis was conducted according to the kit manual, the other two kits also identified the virus at 1 fg/μL of ASSs.

The reaction time of the nine kits was separated into two groups; one group had a running time of around 1 h (50 to 70 min), including the Palm PCR™ ASFV Fast PCR Kit (Ahram Biosystems, Seoul, Republic of Korea), the ASFV-QS (MiCo BioMed, Seongnam-si, Republic of Korea), and the PowerChek™ ASFV Real-time PCR Detection Kit Ver.1.0 (Kogene Biotech, Seoul, Republic of Korea). The other group, including the VDx^®^ ASFV qPCR Kit (Median Diagnostics, Chuncheon-si, Republic of Korea), the Virotype ASFV Real-time PCR Kit (Indical Bioscience, Leipzig, Germany), the ID Gene™ African Swine Fever Duplex (Innovative Diagnostics, Grabels, France), the VetMax™ ASFV Detection Kit (Thermofisher, Waltham, MA, USA), and the Genelix™ ASFV Real-time PCR Detection Kit (Sanigen, Anyang-si, Republic of Korea), had reaction times exceeding 90 min. The comparison results of the reaction time among the finally assessed three kits and the WOAH-recommended methods showed that the three kits were at least 1 h faster than the WOAH methods [23]. Regarding the cost per reaction, the nine kits could be divided into two groups: under KRW 10,000 and over KRW 10,000. The >KRW 10,000 group included three kits from Industrial Bioscience, Innovative Diagnostics, and MiCo BioMed. Thus, cost-effectiveness can be considered an important factor when selecting kits.

## 4. Discussion

Previously, three kits (Virotype ASFV PCR Kit, VetMax™ ASFV Detection Kit, and ID Gene™ African Swine Fever Duplex) were evaluated, including nine commercial PCR kits with WOAH-recommended methods [23]. The difference between the previous study and this study is that internal and synthesized positive controls were not continually amplified with ID Gene™ African Swine Fever Duplex, according to the manufacturer’s instructions. This is because unidentified inhibitors in the PCR were present in the DNA extracted from the food waste samples. The other two kits were successfully assessed under the same conditions. Nevertheless, the three kits, the VDx^®^ ASFV qPCR Kit, the Palm PCR™ ASFV Fast PCR Kit, and the PowerChek™ ASFV Real-time PCR Detection Kit Ver.1.0, evaluated in the previous and this study performed similarly. These tests could be selected or considered because of their high sensitivity, short running time, and cost-effectiveness.

In conclusion, the nine approved ASFV real-time PCR diagnostic kits were successfully assessed, and all kits could detect ASFV. Of the nine kits, some were selected based on their high sensitivity (100 ag/μL), cost-effectiveness (less than KRW 10,000), and short running time (less than 70 min) for ASFV detection in food waste samples. Unfortunately, livestock viruses or viruses from infected samples in the field cannot be tested due to safety and risk prevention considerations. A limitation of our study is that the nine kits were applied only to food waste samples. This study is the first to assess and comprehensively compare approved ASFV diagnostic kits using food waste samples in the Republic of Korea. Overall, this study is expected to provide helpful information and opportunities for selecting kits for the monitoring, early diagnosis, and prevention of the spread of ASF through food waste.

## Figures and Tables

**Figure 1 viruses-16-01627-f001:**
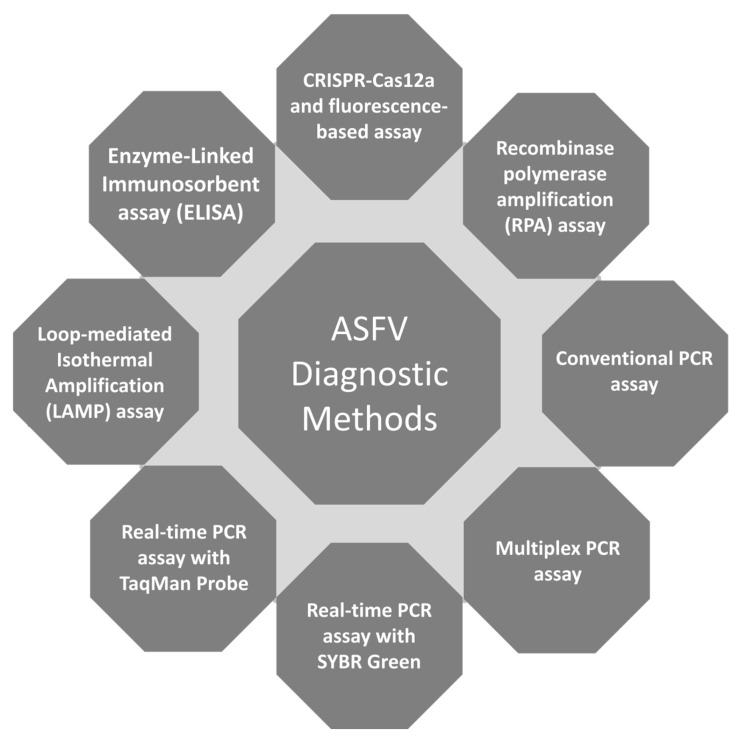
The various diagnostic methods for African swine fever virus (ASFV).

**Figure 2 viruses-16-01627-f002:**
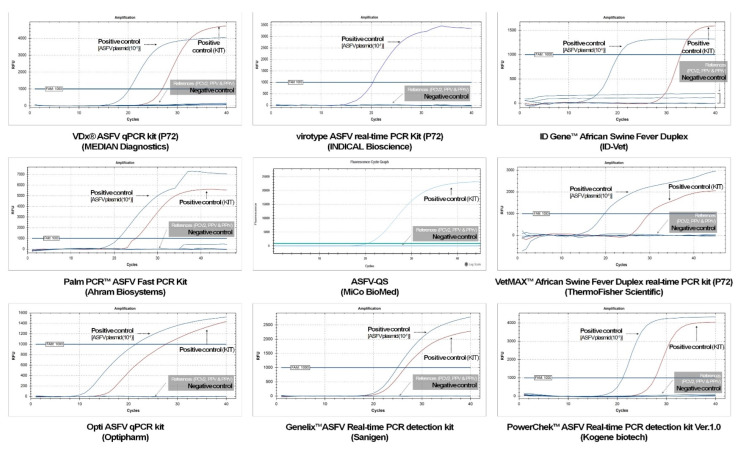
The specificity results of nine real-time PCR kits approved by the Ministry of Agriculture, Food, and Rural Affairs (MAFRA) in the Republic of Korea for detecting African swine fever virus (ASFV). RFU, relative fluorescence unit; PCV2, *Porcine circovirus* 2; PPV, *Porcine Parvovirus*; PPrV, *Porcine Pseudorabies virus*.

**Table 1 viruses-16-01627-t001:** Information on the African swine fever virus real-time PCR diagnostic kits approved by the Ministry of Agriculture, Food, and Rural Affairs (MAFRA) in the Republic of Korea.

No.	Manufacturer	Product		Permit	
		Name	Number of Tests	Number	Date
1	MEDIAN Diagnostics (Chuncheon-si, Republic of Korea)	VDx^®^ ASFV qPCR Kit (P72)	96	121-083	23 November 2018
2	INDICAL Bioscience (Leipzig, Germany)	Virotype ASFV Real-time PCR Kit (P72)	96	150-004	12 July 2019
3	Innovative Diagnostics (ID-Vet; Grabels, France)	ID Gene African Swine Fever Duplex	100	113-007	19 July 2019
4	Ahram Biosystems (Seoul, Republic of Korea)	Palm PCR™ ASFV Fast PCR Kit	96	229-002	12 December 2019
5	MiCo BioMed (Seongnam-si, Republic of Korea)	ASFV-QS	100	081-006	11 February 2020
6	Thermofisher (Waltham, MA, USA)	VetMax™ ASFV Detection Kit	96	128-004	16 March 2020
7	Optipharm (Cheongju-si, Republic of Korea)	Opti ASFV qPCR Kit	96	238-002	8 September 2020
8	Sanigen (Anyang-si, Republic of Korea)	Genelix™ASFV Real-time PCR Detection Kit	48	272-001	8 October 2020
9	Kogene biotech (Seoul, Republic of Korea)	PowerChek™ ASFV Real-time PCR Detection Kit Ver.1.0)	96	136-049	27 October 2020

**Table 2 viruses-16-01627-t002:** Validation results of three real-time PCR kits for analysis of analytical sensitivity using artificially spiked samples (ASSs).

Product and Manufacturer	Experimenter	1000 RFU (Cq Value)								2500 RFU (Cq Value)							
		100 pg	10 pg	1 pg	100 fg	10 fg	1 fg	100 ag	10 ag	100 pg	10 pg	1 pg	100 fg	10 fg	1 fg	100 ag	10 ag
VDx^®^ ASFV qPCR Kit (P72) (MEDIAN Diagnostics, Chuncheon-si, Republic of Korea)	Developer	Detected *				31.47	34.30 ^†^	37.40 ^‡^	N/A **	Detected *			31.64	33.65 ^†^	36.70 ^‡^	N/A **	N/A **
	Analyst 1					33.94 ^†^	37.92 ^‡^	N/A **					33.30 ^†^	36.30 ^‡^	N/A **		
	Analyst 2					34.36 ^†^	37.47 ^‡^						33.60 ^†^	37.49 ^‡^			
	Analyst 3					34.48 ^†,‡^	N/A **						33.48 ^†^	37.31 ^‡^			
	Avg. Cq value					33.56 ^†^	36.56	37.40 ^‡^					33.01 ^†^	36.19	36.70 ^‡^		
	Avg. Sensitivity (Identical/KIT criteria)	10^−5.25^ (Approximately 5.62 fg/uL)/10^−6.00^ (Approximately 1.00 fg/uL)								10^−4.25^ (Approximately 56.23 fg/uL)/10^−5.25^ (Approximately 5.62 fg/uL)							
Palm PCR^™^ ASFV Fast PCR Kit (Ahram Biosystems, Seoul, Republic of Korea)	Developer	Detected *				31.58	34.57 ^†^	38.56^‡^	N/A **	Detected *			30.36	33.65 ^†^	36.79 ^‡^	41.43	N/A **
	Analyst 1					34.84 ^†^	39.87 ^‡^	N/A **					34.60 ^†^	37.43 ^‡^	N/A **	N/A **	
	Analyst 2					34.50 ^†^	37.61 ^‡^						33.40 ^†^	37.16 ^‡^			
	Analyst 3					34.48 ^†,‡^	40.52 **						33.38 ^†^	37.10 ^‡^			
	Avg. Cq value					33.85 ^†^	38.14	38.56 ^‡^					32.94 ^†^	36.34	36.79 ^‡^	41.43	
	Avg. Sensitivity (Identical/KIT criteria)	10^−5.25^ (Approximately 5.62 fg/uL)/10^−6.00^ (Approximately 1.00 fg/uL)								10^−4.25^ (Approximately 56.23 fg/uL)/10^−5.25^ (Approximately 5.62 fg/uL)							
PowerChek^™^ ASFV Real-time PCR Detection Kit (Kogene biotech, Seoul, Republic of Korea)	Developer	Detected *				32.17 ^†^	35.19 ^‡^	N/A **	N/A **	Detected *			30.50	34.28 ^†^	37.28 ^‡^	N/A **	N/A **
	Analyst 1					32.07	34.60 ^†,‡^	39.74					31.51	34.43 ^†^	37.51 ^‡^		
	Analyst 2					31.40	34.99 ^†^	37.23 ^‡^					30.67	33.74 ^†^	37.49 ^‡^		
	Analyst 3					31.36	34.72 ^†^	37.32 ^‡^					30.81 ^†^	35.53 ^‡^	38.83		
	Avg. Cq value					31.75	34.88 ^†,‡^	38.10					30.87	34.50 ^†^	37.78 ^‡^		
	Avg. Sensitivity (Identical/KIT criteria)	10^−5.75^ (Approximately 1.78 fg/uL)/10^−6.50^ (Approximately 316.23 ag/uL)								10^−4.25^ (Approximately 17.78 fg/uL)/10^−5.75^ (Approximately 1.78 fg/uL)							

^†^ Identical Cq criteria, <35.0. ‡ Cq criteria according to kits’ manuals: VDx^®^ ASFV qPCR Kit (P72), <40.0; Palm PCR^™^ ASFV Fast PCR Kit, <40.0; PowerChek^™^ ASFV Real-time PCR Detection Kit, ≦ 38.0. * Detected: positive reaction ** N/A: not applicable.

## Data Availability

All data are included in the manuscript. The ASFV and reference virus genomes were obtained from GenBank, and the GenBank accession number for each sequence is provided in the manuscript. The original contributions presented in the study are included in the article/Appendix A; further inquiries can be directed to the corresponding author/s.

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
