# Peer review of "Assessment of Nine Real-Time PCR Kits for African Swine Fever Virus Approved in Republic of Korea"

_viruses, 2024, doi:10.3390/v16101627_

Round 1

Reviewer 1 Report

Comments and Suggestions for Authors

General comment

This work describes an assessment of diagnostic molecular kits for African swine fever diagnosis in Korea. African swine fever is a very hot topic in the last years and despite ongoing efforts, an effective vaccine is still elusive. An early identification of the virus remain mostly important.   In the abstract, introduction and so on the reference on World Organization for Animal Health is not correct. The introduction must be improved. The author should bee report also the motivations of a comparative study on food waste samples. Which were the charactestics of the samples and why the number is so limited. The criteria selection samples is missed   Lines 39: what do mean four types ……, could the authors explain better. The ASFv is divide in 24 genotypes on the basis of B646L/VP72 and the molecular markers for the genotyping are not only the four regions reported. The authors could improve this sentence. …. Lines 46: …..”few years ago, an ASF vaccine targeting multigene family (MGF) 360-46 505R and CD2v gene deletion emerged as an ASF vaccine candidate”…. The authors could be introduce a reference. Line 54. Many others techniques are developed…, for example….LAMP, Biosensors, etc….The authors could report a complete image of the diagnostic techniques.   Line 62: the number of samples tested is very. limited  

Authors should double-check all references, the figures and the tables.

In conclusion, the article must be improved and after it might have a merit for publication in Viruses after major revision.

Author Response

General comment

This work describes an assessment of diagnostic molecular kits for African swine fever diagnosis in Korea. African swine fever is a very hot topic in the last years and despite ongoing efforts, an effective vaccine is still elusive. An early identification of the virus remain mostly important.

RESPONSE: We agree this comment.

We would like to thank the reviewer for helping us to improve our manuscript. We corrected the manuscript according to the reviewer’s comment.

Comments 1: In the abstract, introduction and so on the reference on World Organization for Animal Health is not correct.

Response 1: We agree this comment. We have revised manuscript where this change can be found - Line 24, 45, 55, 58, 64, 187, 188, 196.

Comments 2: The introduction must be improved. The author should bee report also the motivations of a comparative study on food waste samples. Which were the charactestics of the samples and why the number is so limited. The criteria selection samples is missed.

Response 2: We explain this comment. We have revised manuscript where this change can be found - Line 46-50, Motivations; Line 80-86, Sample size.

Comments 3: Lines 39: what do mean four types ……, could the authors explain better.

The ASFv is divide in 24 genotypes on the basis of B646L/VP72 and the molecular markers for the genotyping are not only the four regions reported.

The authors could improve this sentence. ….

Response 3: We explain this comment. We have revised manuscript where this change can be found - Line 37-42.

Comments 4: Lines 46: …..”few years ago, an ASF vaccine targeting multigene family (MGF) 360-46 505R and CD2v gene deletion emerged as an ASF vaccine candidate”…. The authors could be introduce a reference.

Response 4: We explain this comment. We have revised manuscript where this change can be found - Line 53, reference No 13, 14, 15.

Comments 5: Line 54. Many others techniques are developed…, for example….LAMP, Biosensors, etc….The authors could report a complete image of the diagnostic techniques.

Response 5: We agree this comment. We have revised manuscript where this change can be found - Figure 1 as new figure.

Comments 6: Line 62: the number of samples tested is very. limited

Response 6: We explain this comment. We have revised manuscript where this change can be found - Line 80-86.

Comments 7: Authors should double-check all references, the figures and the tables.

Response 7: All references, the figures and tables have been checked.

In conclusion, the article must be improved and after it might have a merit for publication in Viruses after major revision.

Reviewer 2 Report

Comments and Suggestions for Authors

 This study evaluated nine commercial real-time PCR kits approved in the Republic of Korea using the synthesized 22ASFV plasmid, 20 food waste samples, and artificially spiked samples in analytical sensitivity, analytical specificity, operation time and cost effectiveness. Evaluating the performance of commercial kits using food waste samples is unique aspect and has valuable information for selecting the kits fitting food waste-based surveillance on ASF. However, following comments need to be addressed before further consideration for publication.

Lines 2-3: Suggestion for better title: Assessment of nine real-time PCR kits for African Swine Fever Virus approved in the Republic of Korea

Line 21: Recommend changing ASF into 'further spread of ASF'.

Line 22: Recommend deleting ‘and’.

Lines 24 and many others: reconsider on diagnostic vs analytical in this study: considering this study design, this should be analytical sensitivity and analytical specificity here and other sentences through this manuscript. Diagnostic sensitivity and specificity should be determined based on enough number of clinical specimens from individual pigs.

Lines 29-31: Recommend deleting 'Moreover'. Recommend changing into ' This is the first report on performance comparison of ASFV diagnostic kits approved in the Republic of Korea, providing valuable information for selecting kits for testing with food waste samples.

Lines 38-42: Read carefully on four types vs 25 genotypes vs more recently four new types based on more recent analysis (https://www.mdpi.com/1999-4915/15/11/2246). This sentence is not clear. There rewrite after getting updated understanding on ASFV types.

Line 53: Recommend using ‘Since’ instead of 'After'.

Line 59: This sentence is not clear. Rewrite to make a clearer sentence.

Line 62: Recommend using ‘in’ instead of ‘for’.

Line 69: 20 experimental samples is not enough and proper for diagnostic sensitivity evaluation.

Line 72: Recommend changing into 'To prevent ASFV leakage in testing laboratories and farms,'.

Line 79: Recommend changing into ‘analytical specificity’ instead of ‘specificity’ for further clarity.

Line 88-89: change into ‘Information on African swine fever virus real-time PCR diagnostic kits approved by the Ministry of Agriculture, Food and Rural Affairs (MAFRA) in the Republic of Korea’.

Line 138: analytical specificity, sensitivity looks more accurate considering experimental designs.

Line 142, 145, 152: analytical instead of diagnostic

Lines 169-172: This sentence is not clear. Recommend rewriting.

Line 184: Recommend changing into 'Nevertheless, the three kits, VDx® ASFV qPCR kit, Palm PCR™ ASFV Fast PCR kit, and 185 PowerChek™ ASFV Real-time PCR detection kit Ver.1.0, evaluated in the previous and this study performed similarly. '

Line 189: Change into ‘Of’ instead of ‘Among’.

Lines 189-191: Recommend changing into 'kits, some were selected based on their high sensitivity (100 ag/μL), cost-effectiveness (less than 10,000 won), and short running time (less than 70 min) for ASFV detection in food waste samples.'

Comments on the Quality of English Language

Recommend improving based on my comments and other native speaker's review.

Author Response

Comments and replies

We thank you and the reviewers for your thoughtful suggestions and insights. The manuscript has benefited from these insightful suggestions. We would like to thank you for giving us the opportunity to accept with major revision of our manuscript and the reviewers for their helpful suggestions and comments. The manuscript has been rechecked and the necessary changes have been made in accordance with the reviewers’ suggestions. The responses to all comments have been prepared and attached herewith. The changes in the revised manuscript are given in red color.

Thank you for your decision again.

[Reviewer 2]

This study evaluated nine commercial real-time PCR kits approved in the Republic of Korea using the synthesized 22ASFV plasmid, 20 food waste samples, and artificially spiked samples in analytical sensitivity, analytical specificity, operation time and cost effectiveness. Evaluating the performance of commercial kits using food waste samples is unique aspect and has valuable information for selecting the kits fitting food waste-based surveillance on ASF. However, following comments need to be addressed before further consideration for publication.

RESPONSE: AGREE AND CHANGES MADE

We would like to thank the reviewer for helping us to improve our manuscript. We corrected the manuscript according to the reviewer’s comment.

Comments 1: Lines 2-3: Suggestion for better title: Assessment of nine real-time PCR kits for African Swine Fever Virus approved in the Republic of Korea

Response 1: We agree this comment. We have revised manuscript where this change can be found - Line 2-3.

Comments 2: Line 21: Recommend changing ASF into 'further spread of ASF'.

Response 2: We agree this comment. We have revised manuscript where this change can be found - Line 20.

Comments 3: Line 22: Recommend deleting ‘and’.

Response 3: We agree this comment. We have revised manuscript where this change can be found - Line 21.

Comments 4: Lines 24 and many others: reconsider on diagnostic vs analytical in this study: considering this study design, this should be analytical sensitivity and analytical specificity here and other sentences through this manuscript. Diagnostic sensitivity and specificity should be determined based on enough number of clinical specimens from individual pigs.

Response 4: We agree this comment. We have revised manuscript where this change can be found - line 85, 86, 96, 125, 155, 159, 162, 169, 216.
The title of supplementary figure s1 should be changed diagnostic to analytical.

Comments 5: Lines 29-31: Recommend deleting 'Moreover'. Recommend changing into ' This is the first report on performance comparison of ASFV diagnostic kits approved in the Republic of Korea, providing valuable information for selecting kits for testing with food waste samples.

Response 5: We agree this comment. We have revised manuscript where this change can be found - Line 28-30.

Comments 6: Lines 38-42: Read carefully on four types vs 25 genotypes vs more recently four new types based on more recent analysis (https://www.mdpi.com/1999-4915/15/11/2246). This sentence is not clear. There rewrite after getting updated understanding on ASFV types.

Response 6: We agree this comment. We have revised manuscript where this change can be found - Line 37-42.

Comments 7: Line 53: Recommend using ‘Since’ instead of 'After'.

Response 7: We agree this comment. We have revised manuscript where this change can be found - Line 58.

Comments 8: Line 59: This sentence is not clear. Rewrite to make a clearer sentence.

Response 8: We explain this comment. We have revised manuscript where this change can be found - Line 65-67.

Comments 9: Line 62: Recommend using ‘in’ instead of ‘for’.

Response 9: We agree this comment. We have revised manuscript where this change can be found - Line 69.

Comments 10: Line 69: 20 experimental samples is not enough and proper for diagnostic sensitivity evaluation.

Response 10: We agree this comment. We have revised manuscript where this change can be found - Line 80-86.

Comments 11: Line 72: Recommend changing into 'To prevent ASFV leakage in testing laboratories and farms.

Response 11: We agree this comment. We have revised manuscript where this change can be found - Line 89.

Comments 12: Line 79: Recommend changing into ‘analytical specificity’ instead of ‘specificity’ for further clarity.

Response 12: We agree this comment. We have revised manuscript where this change can be found - Line 96.

Comments 13: Line 88-89: change into ‘Information on African swine fever virus real-time PCR diagnostic kits approved by the Ministry of Agriculture, Food and Rural Affairs (MAFRA) in the Republic of Korea’.

Response 13: We agree this comment. We have revised manuscript where this change can be found - Line 105-106.

Comments 14: Line 138: analytical specificity, sensitivity looks more accurate considering experimental designs.

Response 14: We agree this comment. We have revised manuscript where this change can be found - Line 155.

Comments 15: Line 142, 145, 152: analytical instead of diagnostic.

Response 15: We agree this comment. We have revised manuscript where this change can be found - Line 159, 162, 169.

Comments 16: Lines 169-172: This sentence is not clear. Recommend rewriting.

Response 16: We agree this comment. We have revised manuscript where this change can be found - Line 186-189.

Comments 17: Line 184: Recommend changing into 'Nevertheless, the three kits, VDx® ASFV qPCR kit, Palm PCR™ ASFV Fast PCR kit, and 185 PowerChek™ ASFV Real-time PCR detection kit Ver.1.0, evaluated in the previous and this study performed similarly.

Response 17: We agree this comment. We have revised manuscript where this change can be found - Line 201-203.

Comments 18: Line 189: Change into ‘Of’ instead of ‘Among’.

Response 18: We agree this comment. We have revised manuscript where this change can be found - Line 206.

Comments 19: Lines 189-191: Recommend changing into 'kits, some were selected based on their high sensitivity (100 ag/μL), cost-effectiveness (less than 10,000 won), and short running time (less than 70 min) for ASFV detection in food waste samples.

Response 19: We agree this comment. We have revised manuscript where this change can be found - Line 206-208.

Comments 20: Comments on the Quality of English Language

Recommend improving based on my comments and other native speaker's review.

Response 20: This manuscript has been edited by the Editage for language and grammar accuracy. The certificate is attached.

Round 2

Reviewer 1 Report

Comments and Suggestions for Authors

Journal: Viruses_Section Animal Viruses-3208148

Assessment of Nine Approved Diagnostic Kits in the Republic 2 of Korea Based on Real-time PCR for African Swine Fever Virus citation.

Siwon Lee 1, †, Tae Uk Han 2, †, Jin-Ho Kim 3, 4, *

General comment

This work describes an assessment of diagnostic molecular kits for African swine fever diagnosis in Korea. African swine fever is a very hot topic in the last years and despite ongoing efforts, an effective vaccine is still elusive. An early identification of the virus remain mostly important.

The manuscript was revised and was integrated also taking into account the suggestions of the reviewers.

The article, have a merit for publication in Viruses in this form.